# A Model of Threats to the Confidentiality of Information Processed in Cyberspace Based on the Information Flows Model

**Egoshin N. S., Konev A. A.** 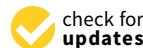 **and Shelupanov A. A. ***

Department of Complex Information Security of Computer Systems, Tomsk State University of Control Systems and Radioelectronics, 40 Lenina Prospect, 634050 Tomsk, Russia; ens@csp.tusur.ru (E.N.S.); kaa1@keva.tusur.ru (K.A.A.)

*  Correspondence: saa@tusur.ru; Tel.: +7-(3822)-7015-29

**Abstract:** This article covers one of the fundamental problems of information security—building a threat model. The article discusses a new method for identifying typical threats to information confidentiality based on the information flow model. The threat model is based on the description of the system. An incorrect description of the system leads to the formation of an incorrect threat model. A review of the subject area revealed several approaches used to describe the system in terms of circulating information flows. Each of these approaches has its own pros and cons. The model of information flows proposed in this work reduces the description of any information system to an eight-digit alphabet. Analysis of the structure of the elementary information flow identified four typical threats to confidentiality, the Cartesian product of a set of threats and a set of streams is a complete model of typical threats to the confidentiality of information processed in cyberspace.

**Keywords:** information security; confidentiality; threat model; information flow model

## 1. Introduction

With the development and formation of the information society, the problem of ensuring information security is becoming more and more urgent. All modern organizations strive to increase the integration of information technologies in their areas of activity, because this allows you to move to a qualitatively new level of storage, processing and transmission of information.

Unfortunately, due to the same integration of information technologies not only into business, but also into private life, many ordinary people have lost their awareness of the conceptual difference between the real and virtual environment of interaction with other people. Very often there are situations when an unprepared person does not even understand that the information he is sharing applies not only to him. Without realizing it, he may violate the confidentiality regime of trade secrets directly or give out information that may indirectly lead to this.

If we approach the problem from the point of view of systems analysis, then we can try to abstract and generalize to a single concept all the ways of interaction between any objects that store, process and transmit information. Any case of information transfer can be represented as a kind of information flow between the source and the recipient. Using this concept, the whole essence of information protection can be reduced to one goal—it is necessary to ensure the safety of all elements of any elementary information flow at any moment of time.

In fact, any information security breach incident can be described as follows: any unauthorized action in an information system is the emergence of an unvalidated information flow.

At its core, the process of determining a complete list of permitted information flows in the system is the usual delimitation of access to information.

However, as in all rapidly developing areas of human knowledge, there is no proper unification in the field of information security. Legally established methods are often of an exclusively recommendatory nature and at the same time contain very lengthy formulations. In this regard, information security specialists are forced to develop their own local regulations. It is obvious that in such a situation the expert's professionalism and his subjective opinion significantly affect the final result.

However, that is not all. Over the past decades, the integration of network technologies in all spheres of life has grown exponentially. Distributed systems have long become a completely common thing, no one can be surprised by smart things and IoT technology in general.

In this regard, a new urgent problem has appeared. Moreover, this importance has increased not from the side of functionality, but from the point of view of information security. In the abstract case, from the point of view of the system's logic, nothing has changed—we still connect two devices with a communication line and transfer information—but everything is not so simple. Some ten years ago, in order to send information from the phone to the computer, we connected the cable and did not think about anything else, because here it is, the cable lies in plain sight, no one touches it, or tries to cut it, or removes the lead and replaces it with their own. However, now the situation has changed. It is much faster and easier to use wireless communication channels, not only local, but even cloud technologies. In view of the rapid development of technologies and, accordingly, their speed, less and less attention is being paid to optimizing the use of resources. As a result, instead of going 40 cm over a personal cable, information travels thousands of kilometers over backbone networks, passing through dozens of servers. Information security has become highly dependent on the reliability of network service providers. It has become much more difficult to determine probable threats, because now an information security specialist is simply not able to study the internal structure of the system [1].

It is important to understand that the emergence of new technologies not only gives rise to new methods of attacks, but also expands the existing list of threats, and, as you know, each threat can be carried out by a large number of different attacks. The emergence of new technologies non-linearly reduces the level of security of existing systems. In this regard, the need to form a complete list of information threats comes to the fore, but this problem does not have a simple solution. What can we say about ordinary users, even if information security specialists cannot always correctly compile a complete list of all possible threats. To solve this problem, various threat models are created, which are based on all kinds of mathematical tools and information models.

It is worth saying that the multiplicity of different models is due not only to the difference in the views of researchers and their approaches to solving the problem. The solutions used to protect information depend on the aspect of information security [2]. We cannot use the same models to protect confidentiality and integrity or availability, just as we cannot use the same models to predict attacks on information and on a system, since objects are fundamentally different from each other. Yes, the models can have the same apparatus at their core, but their final mechanism will be different [3].

From all of the above, the following theses follow:

— when determining the list of threats to confidentiality, integrity and availability of information, different threat models should be used;
— it is necessary to exercise strict control over information transmission channels;
— the process of forming a threat model must consider not only the nodes of the system, but also the channels, otherwise such a model will never reach completeness.

This work provides a study of the current state of the subject area, and also proposes its own version of the solution of the given tasks. In addition to the introduction, the article contains four main sections. In the first section, an overview of the subject area was carried out. The second section contains a description of the information flow model and an example of its application. The third section is devoted directly to the model of threats to the confidentiality of information, as well as the

justification of its completeness. In the fourth section, the concretization and typification of the selected sets of threats and comparison with the most popular analogue, the STRIDE model, are given.

## 2. Background and Related Work

Information flow has been studied since the mid 1970s, although research has mainly been concerned with stateful, temporal computations [4]. In various scientific works, the topic of the need to apply the model of information flows when solving information security issues has been raised more than once, for example, when building a threat model or forming a policy of access control.

Information flow theory is applicable to a wide range of types of systems: cyber-physical (CPS), telemedicine systems, SCADA, IoT, software development systems. Let's take a closer look at all types of systems.

Speaking of cyber-physical systems, the following publications can be distinguished: in [5], the authors report that, information flow is a fundamental concept underlying the security of a system and confidentiality of information in a system can be breached through unrestricted information flow, this statement coincides with the ideas of the authors expressed in the Introduction to the current article; in turn, the authors of [6] do not pay so much attention to specific information flows, speaking more about physical flows, but at the same time mentioning that access control and information flow-based policies for CPS security should be analyzed.

Telemedicine systems are becoming more and more popular, and there are also publications on this topic with similar views: if the authors of [7] and [8] propose to use the Flow Diagrams via STRIDE or DREAD methodology, the authors of [9] do not introduce a separate concept of information flow, but The Network is said to be an important part of the system along with Clients and Servers.

In [10], the information flow is also mentioned only once in the context of the fact that it can be the object of an attack; however, in the developed methodology, only threats aimed at system elements are determined, while the flow itself is not classified as such.

Increasingly, threat models are being applied early in software development. There are works that adhere to the STRIDE methodology, such as [11–13], but at the same time, there are works that completely ignore this topic, for example [14].

Having started talking about software, in no case can you ignore SCADA systems, because by the number of information transmission channels they are likely to bypass all other systems, and already a violation of the information security mode in these channels can lead to catastrophic consequences (economic losses or even man-made disasters). The authors of articles [15,16] use approximately the same approach: their models contain threats directed at the channel, but the channel itself is not considered as a significant part of the system, and without a comprehensive analysis it is difficult to speak about the completeness of the model itself.

Moving on to the issue of large and distributed systems, it is imperative to mention things that have become commonplace, such as cloud technologies and IoT. As expected, as the system grows, the importance of communication lines also increases. For an IoT system, an information transmission channel is no longer just an obligatory element, but becomes a full-fledged working unit [17]. Accordingly, the works on this topic have great unanimity. The authors of [18], although they do not say anything about information flows, nevertheless believe that identification, assessment, and mitigation of risk will be more difficult and complex for cloud computing, mobile device toting, consumerized enterprise. The authors of works [19–22] indicate the importance of accounting for information flows and suggest using the DFD and STRIDE methodology.

If we move away from the applied solutions, which were discussed above, and look at the subject area as a whole, it will become clear that publications on this topic have been coming out with enviable regularity for more than a decade.

In addition to the above works, there are also works that relate to unique subject areas or even have a general purpose, but nevertheless one way or another mention information flows when building a threat model. Some of them [23–25] even use a similar term-flow. In addition, in several works [26–34],

the term network is used, although from the context it becomes obvious that this is not a network connection, but an information flow.

These concepts must be strictly delineated, since their mixing and substitution only cause great confusion. Not every information flow is implemented by a network, but all networks implement flows. A network connection is only a particular kind of information flow. The very concept of a stream is much more extensive and defines all possible channels for transmitting information.

When considering the scheme of information flows, most of the works use the DFD methodology; however, it has several critical drawbacks:

— the model has two separate notations for building internal and external interaction schemes;
— the model does not describe the channels of information transmission and the resulting information flows.

The paper [35] proposes the use of Hidden Markov Chains for asymmetric threat modeling. However, this approach is not correct, since it is more expedient to use Markov chains when defining attacks, which, in contrast to threats, are probabilistic in nature.

At the end of the literature review, it is worth mentioning a very interesting solution to the problem, namely, the use of Petri nets [36]. Certainly, Petri nets are convenient to use when modeling discrete automata, both finite and infinite. However, in the context of the current research, this makes no sense, since Petri nets make it possible to describe the process of information transfer, or rather the very fact of information transfer from one vertex to a channel and further, but do not allow the separate description of the information transfer channel. It makes no sense to model the system at different locations of information in it. A higher level of abstraction is required, when information can be found both in all elements at the same time, or in any other combination, up to its complete absence.

Summarizing the overview, it is necessary to outline one important detail. The problem is that STRIDE does not simulate threats, but attacks. These terms are undoubtedly related, but they still should not be confused. Threats are broader than attacks, and each threat can be implemented by many different attacks. Comprehensive measures to combat threats are of a preventive nature. Threat overlap provides protection against a large layer of attacks. Therefore, the formation of a threat model is of primary importance.

## 3. Protected Object Structure

To form a model of information flows, it is proposed to use the concepts of graph theory. A detailed rationale and implementation of this approach follows.

Any information exchange scheme can be represented as a collection of elementary information flows. Elementary information flow includes three elements: transmitter, information transmission channel, receiver.

This concept can be clearly demonstrated if we apply graph theory. Let us introduce the following notation: V is a set of information carriers (a set of graph vertices), E is a set of information transmission channels (a set of graph edges). Comparing any two elements from V and one from E, we get an elementary information flow in the form of an undirected graph with two vertices (Figure 1).

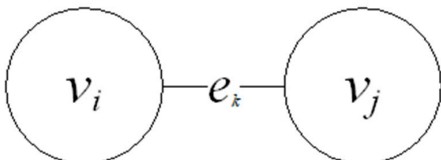

**Figure 1.** Elementary information flow.

Using the graph theory notation, we describe the above information flow:

$$g = (v_i, e_k, v_j), \tag{1}$$

where $v_i$, $v_j$ are possible carriers of information; $e_k$ is a possible communication channel.

Each elementary information flow is a symmetric structure, it means that the communication channel is bidirectional. Considering the specifics of the study, namely the work with the organization's electronic information resources, certain sets were compiled.

The set of information carriers was divided into three subsets and took the form

$$V = \{V_1, V_2, V_3\}, \tag{2}$$

where $V_1$ is a set of users; $V_2$ is a set of software tools; $V_3$ is a set of electronic resources.

It is important to make one clarification. Obviously, the user cannot interact with electronic data or software directly; moreover, we are not interested in how exactly the user assimilates information, be it visually, acoustically or otherwise. The main thing is to trace the path of information from the storage device to the moment when the user gets the opportunity to directly exercise his right of access. Therefore, in the context of this study, we will assume that the user interacts with the software by means of the operating system. This means that we will consider operating system modules as communication channels between the user and the software: drivers for input-output devices, shared memory, etc.

To concretize the set of remoting channels, we turn to the OSI model. Having studied all its levels and the protocols used, you can divide them into specific groups. The first classification is obvious and follows from the definition of the levels themselves—communication channels can be divided into local and remote. The second classification follows from the peculiarities of the operation of protocols and devices at different levels—channels can be divided into operating in a virtual and electromagnetic environment.

If we combine these two classifications, then the set of information transmission channels will take the following form:

$$E = \{e_1, e_2, e_3, e_4\}, \tag{3}$$

where $e_1$ is a transmission channel in the electromagnetic environment; $e_2$ is a transmission channel in the virtual environment; $e_3$ is a remote transmission channel in the electromagnetic environment; $e_4$ is a remote transmission channel in the virtual environment.

It is worth noting that, since this classification is not tied to the OSI model, but only uses it to determine the channels for transmitting information, it also considers the ability to read information from local carrier (hard disk or USB media), the only difference is the technologies used; drivers are used for communication, not protocols.

Now it is necessary to get a complete picture. Having an extended set V and a concretized set E, it is possible to construct a set of all elementary information flows G. For this, it is necessary to indicate some restrictions:

— an element of the set $V_1$ cannot refer to another element of this set (in fact, it can, but since in the framework of this study we consider only interactions by means of a computer system, such flows are not considered);
— an element of the set $V_3$ cannot refer to another element of this set;
— an element of the set $V_1$ cannot directly access an element of the set $V_3$ and vice versa;
— channels for remote transmission of information are available only when an element of the set $V_2$ is connected with an element of the same set.

Considering all of the above, the set of all elementary streams will have the following form:

$$G = \{g_1, g_2, g_3, g_4, g_5, g_6, g_7, g_8\}, \tag{4}$$

where $g_1 = \{V_1, e_1, V_2\}$; $g_2 = \{V_1, e_2, V_2\}$; $g_3 = \{V_2, e_1, V_2\}$; $g_4 = \{V_2, e_2, V_2\}$; $g_5 = \{V_2, e_3, V_2\}$; $g_6 = \{V_2, e_4, V_2\}$; $g_7 = \{V_2, e_1, V_3\}$; $g_8 = \{V_2, e_2, V_3\}$.

The result of combining all the above graphs will be an undirected multiplicative graph (Figure 2), which will represent a model of information flows when accessing electronic information resources. It should be noted that the connections between each pair of vertices are symmetrical, i.e., bidirectional. When defining each individual elementary information stream, the direction of movement of information in it does not matter, since we will only be interested in the fact of establishing communication and transferring information. Direction of movement can play a role in identifying threats to information, but this will be discussed later.

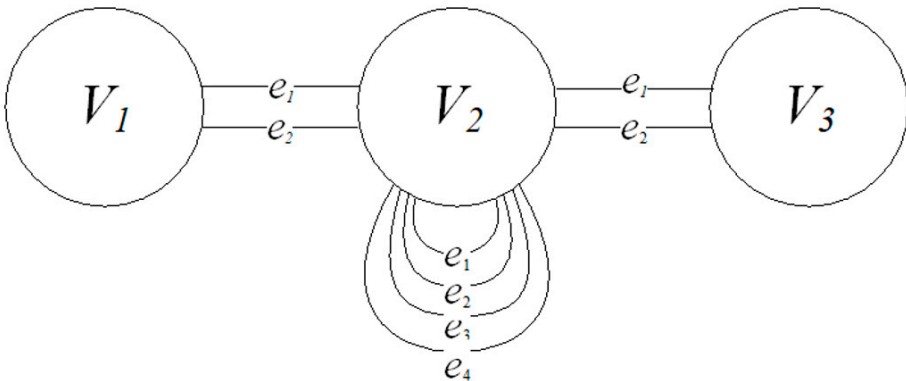

**Figure 2.** Model of information flows in the form of a multiplicative graph.

The developed model makes it possible to construct a scheme of information flows, which, in addition to the allowed flows, will include all possible occurrences of prohibited ones. Knowing the complete list of permitted and prohibited flows, one can not only realize access control, but also try at an early stage to predict and/or track unauthorized actions of a possible internal violator. This approach can potentially turn out to be highly effective when using a scheme of information flows together with a probabilistic model of attacks, but this area of work is beyond the scope of the proposed study.

On the other hand, if we consider only permitted flows, then, having a general model, it will not be difficult to determine the list of typical threats for any of the elements of the scheme, which will make it possible to compile a complete list of threats to the information processed in the system.

Let us demonstrate the application of the information flow model using the example of the process of sending/receiving email.

We will assume that the correspondence is carried out between two users from their mobile devices, which are connected to the Internet by a wireless connection (the general scheme of the described process is shown in Figure 3).

General description of the process of sending/receiving email:

— the first user sends a letter to the mail server of his provider;
— the postal provider sends a letter to the recipient's provider's server;
— the recipient's provider server sends the letter to the recipient.

To compile a complete list of information flows, let us add a few more explanations to the description of the process:

— information on the user's device is not generated by itself (although this situation is also possible), we will assume that in our case, the letter has only a text component and is typed by the user;
— Mail Transfer Agent (MTA) does not store messages, it only transfers them to the Mail Delivery Agent (MDA); situations are possible when MTA and MDA are one device, but in this example, we will distinguish them for greater clarity of the model;
— MDA (as with MTA) is a kind of hardware and software complex that consists of a message sending software and a data store; in our case, the interaction between the mobile mail client and the server software, which already interacts with the server storage, is carried out;

—　we will assume that the interaction between the recipient's mail client and MDA is carried out over the IMAP protocol.

From the above, we get the following list of information flows:

1. Sender—Mail User Agent (MUA);
2. MUA—MTA;
3. MTA—MDA;
4. MDA—Server storage;
5. Server storage—MDA;
6. MDA—MUA;
7. MUA—Recipient.

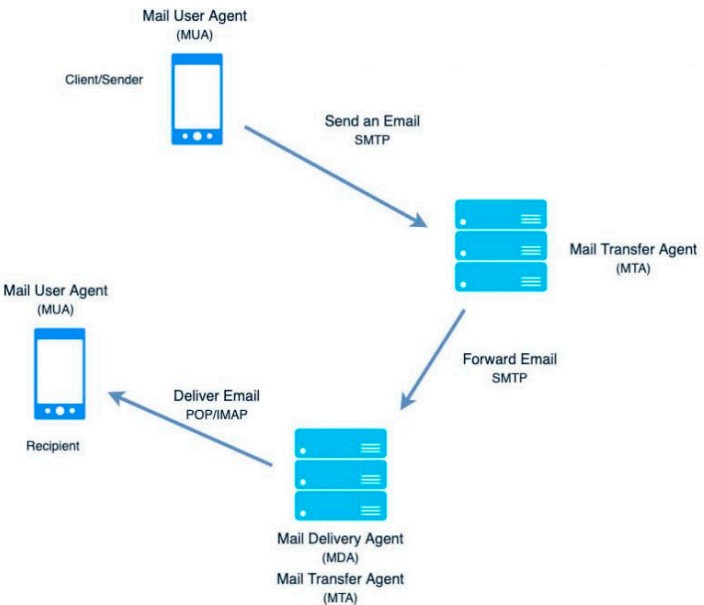

**Figure 3.** General diagram of the email transmission process.

Now let us construct a complete list of elementary information flows from this list of information flows, considering that MDA and MTA are different devices. However, in addition, we will assume that from the MDA side, one piece of software is responsible for the processes of receiving, writing to the storage, reading from it and sending it to the MUA. Each stream is divided into two elementary ones, since the model implies dividing the data transmission channel into electromagnetic and virtual. Additionally, we introduce the designations of all participants in the process according to the information flow model.

User set:

$$V_1 = \{v_{11}, v_{12}\}, \tag{5}$$

where $v_{1,1}$ is a sender; $v_{1,2}$ is a recipient.

Software set:

$$V_2 = \{v_{2,1}, v_{2,2}, v_{2,3}, v_{2,4}\}, \tag{6}$$

where $v_{2,1}$ is a sender MUA software; $v_{2,2}$ is an MTA software; $v_{2,3}$ is an MDA software; $v_{2,4}$ is a recipient MUA software.

Set of repositories of electronic information:

$$V_3 = \{v_{3,1}\}, \tag{7}$$

where $v_{3,1}$ is server storage (database).

The resulting set of elementary information streams will be as follows:

$$S = \{s_1, s_2, s_3, s_4, s_5, s_6, s_7, s_8, s_9, s_{10}, s_{11}, s_{12}, s_{13}, s_{14}\}, \tag{8}$$

where $s_1 = (v_{1,1}, e_1, v_{2,1})$; $s_2 = (v_{1,1}, e_2, v_{2,1})$; $s_3 = (v_{2,1}, e_3, v_{2,2})$; $s_4 = (v_{2,1}, e_4, v_{2,2})$; $s_5 = (v_{2,2}, e_3, v_{2,3})$; $s_6 = (v_{2,2}, e_3, v_{2,3})$; $s_7 = (v_{2,3}, e_1, v_{3,1})$; $s_8 = (v_{2,3}, e_2, v_{3,1})$; $s_9 = (v_{3,1}, e_1, v_{2,3})$; $s_{10} = (v_{3,1}, e_2, v_{2,3})$; $s_{11} = (v_{2,3}, e_3, v_{2,4})$; $s_{12} = (v_{2,3}, e_4, v_{2,4})$; $s_{13} = (v_{2,4}, e_1, v_{1,2})$; $s_{14} = (v_{2,4}, e_2, v_{1,2})$.

As you can see, the entire process of information transmission can be described using a set of elementary information streams, which together form a complete scheme of information flows.

Therefore, using the process of sending/receiving email as an example, the work of the information flow model was illustrated. This analysis shows that the use of the model makes it possible to split any information transfer process into a finite set of elementary information flows, while the only difficulty lies in the correct description of the sets of system elements. The more fully and accurately the sets of elements are described, the more detailed the flow diagram will be. This implies another thesis related to the application of the developed model: an expert is in any case subjective, it is not possible to completely get rid of subjectivity, but you can shift it to a relatively less critical side—a correct and complete description of the system considering all its elements instead of haphazard definition probable threats.

## 4. Information Confidentiality Threat Model

The issue of information security implies an integrated approach. It is necessary to touch upon the maximum possible number of aspects in the field of information protection, in particular, to define a complete list of threats and in the future to use this list of threats in relation to a specific system. It is the completeness of the list of threats that is important, since in the absence of any element, the probability of compromising information and/or the system increases sharply. By definition, threat modeling is a risk management strategy to proactively secure [37]. Thus, the formation of a threat model capable of providing a complete list of threats is a top priority for information security. Threat models should be the starting point for assessing risks and designing future security systems for computer and information systems.

The main problem is that, today, all available models are very conditional. There is no single principle for building a threat model. There are many approaches, and each of them is interpreted in its own way: the lack of a clear concept of the "threat model", the striking difference in the structures and principles of the models' functioning, the ways of using the model, the redundancy of the model in the form of merging with the intruder's model, and much more. The presence in the aggregate of the considered shortcomings negatively affects the effectiveness of the expert's work with the model itself and the final result, due to the lack of standardized final assessments of one threat model relative to another.

As a result of all of the above, the task was set to create its own model of information threats.

The principle of constructing a threat model is based on the developed model of information flows, namely, on the concept of an elementary information flow. Let us turn again to the definition of the elementary information flow, which is described by the formula:

$$g = (v_i, e_k, v_j), \tag{9}$$

where $v_i$, $v_j$ are possible carriers of information; $e_k$ is a possible communication channel.

It is obvious that an information transmission channel is not some abstract object, but a very real element of the system, which has some physical and/or virtual properties. This means that it can be accessed in the same way as the other two flow elements.

Let us try to define and classify the types of access in general terms. Unauthorized access to information is access to protected information in violation of established rights and/or access rules

leading to leakage, distortion, forgery, destruction, blocking access to information, as well as loss, destruction or malfunction of the information carrier.

The very definition of unauthorized access implies the appearance in the system of a new element that will provide this access.

Using the previously given notation, this situation can be depicted as follows (Figure 4).

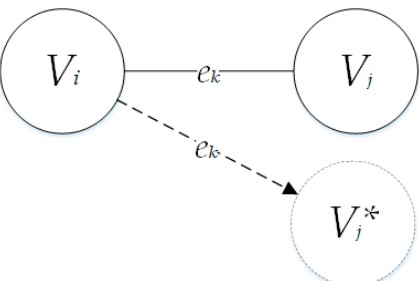

**Figure 4.** Occurrence of an unauthorized element $V_j$ * that receives information from the element $V_i$.

A similar situation is possible for any element of the information flow. By analogy with the situation described above (Figure 4), access can be performed both to the element $V_j$ (Figure 5) and to $e_k$ (Figure 6).

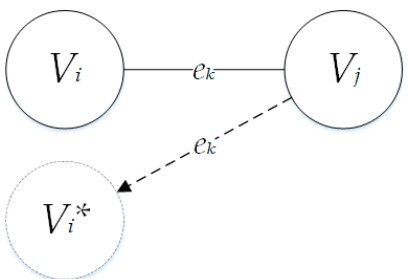

**Figure 5.** Occurrence of an unauthorized element $V_i$ * that receives information from the element $V_j$.

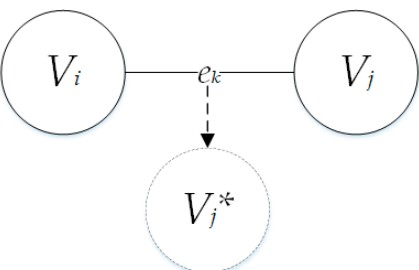

**Figure 6.** Occurrence of an unauthorized element $V_j$ * that receives information from the element $e_k$.

If we talk exclusively about the confidentiality of information, then by definition its violation does not imply a violation of the integrity or availability, although it can lead to this. If we go back to the concept of information flow, it becomes obvious that confidentiality is violated when any of its elements is replaced, i.e., the following cases are possible:

— substitution of any of the two vertices;
— substitution of the channel.

In this case, situations are possible when several elements will be compromised at once. Now, knowing the total number of elements and the number of states of these elements, we can calculate the total number of states of the elementary information stream.

To do this, let us apply the formula for calculating the cardinality of a set:

$$N = p^i, \tag{10}$$

where $p$ is the number of states for an item; $i$ is the number of items.

In our case, $p = 2$, because any flow element can have two states—compromised or not—and $i = 3$, since an elementary information stream consists of three elements. As a result, the total number of elementary information stream state elements tied to compromise will be eight. All possible combinations are presented in Table 1. The table cells contain information about the status of the element being compromised, where "1"—the element is compromised; and "0"—not compromised.

**Table 1.** List of information flow states.

| $V_i$ | $e_k$ | $V_j$ |
|-------|-------|-------|
| 0 | 0 | 0 |
| 0 | 0 | 1 |
| 0 | 1 | 0 |
| 0 | 1 | 1 |
| 1 | 0 | 0 |
| 1 | 0 | 1 |
| 1 | 1 | 0 |
| 1 | 1 | 1 |

However, when building a threat model, there is no need to consider composite layout options, since this approach will lead to a high level of duplication of various threats; therefore, it will be sufficient to consider only four basic states: either one of the elements ($V_i$, $V_j$, $e_k$) is compromised, or no element is compromised.

It is necessary to separately analyze the situation when none of the elements of the system are compromised. The fact is that in addition to a simple substitution, a situation of "eavesdropping" is possible, i.e., access to the information stored in it from outside the controlled area. However, "eavesdropping" will no longer apply to all three elements, since node tracking implies either the introduction into an already existing information transmission channel, which is identical to listening to the channel, or the emergence of a new unauthorized one, which coincides with the channel substitution, and there is still an option in which the entire system is compromised.

Thus, having analyzed all possible types of interference in the information flow, it is possible to build a complete set of typical threats to the confidentiality of information.

Let us describe a set of privacy threats:

$$K = \{k_1, k_2, k_3, k_4\}, \tag{11}$$

where $k_1$, $k_2$, $k_3$, $k_4$ are typical threats to information confidentiality.

Let us analyze each of them in more detail:

$k_1$—substitution of the receiver $V_i$ (receiving of protected information by an unauthorized element $V_i$ *);

$k_2$—substitution of the receiver $V_j$ (receiving of protected information by an unauthorized element $V_j$ *);

$k_3$—presence of an unauthorized channel $e_k$ (substitution of the channel for unauthorized one);

$k_4$—control of the $e_k$ channel (receiving of information by an unauthorized person from outside the authorized zone).

Let us look at the email example again and see how these four common threats can be applied to it.

First, consider the first flow $s_1 = (v_{11}, e_1, v_{21})$. Recall: $v_{11}$ is the sender user, $e_1$ is the electromagnetic channel, $v_{21}$ is the Mail User Agent (MUA).

Let us apply each of the four threats to this flow. Again, remember that the connecting channel in the flow is symmetrical and accordingly bidirectional.

When the $k_1$ threat is implemented, the user ($v_{11}$) is substituted by an unauthorized user ($v_{11}$ *), as a result of which this element can gain access to confidential information. An example of the realization of a threat is the transfer of the phone to a third party. An unauthorized user can familiarize himself with the information in the information transmission channel (typed text in the input interface) or, using the mail client, read other letters sent or received earlier.

When the $k_2$ threat is implemented, the mail client ($v_{21}$) is replaced with unauthorized software ($v_{21}$ *), as a result of which an authorized user ($v_{11}$) can transfer confidential information to an outside software tool. An example would be installing an application from an unverified source.

When the $k_3$ threat is implemented, the communication channel ($e_1$) is replaced. In this case, the communication channel is the input/output device, which, given the current realities, is most likely a touch screen. An example of the implementation of the $k_3$ threat is the substitution of an input/output device: during the repair, an additional touch panel could be imposed on the touchscreen, similar to overhead keyboards for ATMs.

When the $k_4$ threat is implemented, there is no direct intervention in the information channel, there is no substitution of any of the nodes or channel. Information is accessed from outside. An example of implementation is the installation of embedded malicious software or hardware. In fact, built-in malicious hardware does not replace any of the elements, it does not even interfere with their normal operation. Moreover, remote peeping can serve as an example of implementation; none of the elements of the elementary information stream has been compromised, and still there is information leakage outside the system.

A similar selection of examples of the implementation of threats to the confidentiality of information can be selected for another stream and its elements; however, we will not give these analyzes here, since this process is monotonous and, at the same time, will no longer allow to reflect more deeply the essence of the threat model operation. In addition, now, in order to put a semantic point in the study of the example, we note the following: it was possible to notice that in the example under consideration there are fourteen information flows, while in the model of information flows there are only eight of them. The fact is that the model is abstract by definition, and each of its information flow can be represented by an infinite number of examples from real life.

Taking a closer look at the elements of the set S, one can pay attention to their similarity with the elements of the set G.

Considering the symmetry of the flows and the belonging of the vertices to the same sets, we can uniquely associate all elements of the set S with elements of the set G:

— $g_1 \sim s_1, s_{13}$;
— $g_2 \sim s_2, s_{14}$;
— $g_5 \sim s_3, s_5, s_6, s_{11}$;
— $g_6 \sim s_4, s_{12}$;
— $g_7 \sim s_7, s_9$;
— $g_8 \sim s_8, s_{10}$.

It follows from this that any process of information transmission in the system cannot only be described by a set of elementary information streams, but reduced to a finite set of such streams. In addition, recall that the cardinality of this set is only eight, i.e., all channels of any electronic information transmission system can be described using an eight-digit alphabet. Undoubtedly, such a description will not be enough for most needs associated with the use of information systems; however, in the context of information protection and defining a list of threats to information, this will be more than enough.

It should be added that in the implementation of any of the threats, of course, there is a likelihood of further intervention in the system with a subsequent violation of integrity and/or availability, but

this work implies the identification of only the initial threats, and not the identification of risks or of a cascade violation of the information security regime. Identifying only priority threats is not a limitation of the model, but a reflection of its preventive nature.

Now back again to the set of elementary information streams and the set of privacy threats. Knowing that both of these sets are finite, we can apply each of the threats to each flow and get a new set that will consist of all combinations of threats and flows, i.e., be their Cartesian product.

$$
\begin{aligned}
\text{G} \times \text{K} = \{ & g_1k_1, g_1k_2, g_1k_3, g_1k_4, \\
& g_2k_1, g_2k_2, g_2k_3, g_2k_4, \\
& g_3k_1, g_3k_2, g_3k_3, g_3k_4, \\
& g_4k_1, g_4k_2, g_4k_3, g_4k_4, \\
& g_5k_1, g_5k_2, g_5k_3, g_5k_4, \\
& g_6k_1, g_6k_2, g_6k_3, g_6k_4, \\
& g_7k_1, g_7k_2, g_7k_3, g_7k_4, \\
& g_8k_1, g_8k_2, g_8k_3, g_8k_4 \}
\end{aligned}
\tag{12}
$$

Let us calculate the cardinality of the final set:

$$
|\text{G} \times \text{K}| = |\text{G}| * |\text{K}| = 8 * 4 = 32
\tag{13}
$$

It follows from this that by analogy with the description of the set of information flows, we can reduce the set of threats to the confidentiality of information in the system to a finite set of typical threats, the power of which is equal to thirty-two.

## 5. Classification of Typical Threats by Access Object

Now we classify and give a brief description of the defined typical threats. For convenience and readability, the set of typical threats was grouped according to their belonging to information flows from the set G. The following eight tables (Tables 2–9) show the grouping and characteristics of the analyzed typical threats.

In Tables 4–7, the first two typical threats coincide in pairs, since these streams are symmetric.

**Table 2.** Typical threats to information flow $g_1 = (V_1, e_1, V_2)$.

|  | Threat Description | Example of a Threat Implementation | Item Name |
|---|---|---|---|
| $k_1$ | Transfer of information by an authorized process to an unauthorized person | User account spoofing | $g_1k_1$ |
| $k_2$ | Acceptance of information from an authorized person by an unauthorized process | Using an unauthorized device | $g_1k_2$ |
| $k_3$ | Transmission of information through an unauthorized channel in an electromagnetic environment | Use of unauthorized or compromised input/output devices | $g_1k_3$ |
| $k_4$ | Obtaining information from outside the authorized area | Retrieving from TEMPEST information about elements of hardware interfaces | $g_1k_4$ |

**Table 3.** Typical threats to information flow $g_2 = (V_1, e_2, V_2)$.

|  | Threat Description | Example of a Threat Implementation | Item Name |
|---|---|---|---|
| $k_1$ | Transfer of information by an authorized process to an unauthorized person | User account elevation | $g_2k_1$ |
| $k_2$ | Acceptance of information from an authorized person by an unauthorized process | Using an unauthorized or compromised application | $g_2k_2$ |

**Table 3.** *Cont.*

| | Threat Description | Example of a Threat Implementation | Item Name |
|---|---|---|---|
| $k_3$ | Transmission of information through an unauthorized channel in a virtual environment | Use of unauthorized or compromised input/output device drivers, audio and video drivers | $g_2k_3$ |
| $k_4$ | Obtaining information from outside the authorized area | Reading information from the clipboard | $g_2k_4$ |

**Table 4.** Typical threats to information flow $g_3 = (V_2, e_1, V_2)$.

| | Threat Description | Example of a Threat Implementation | Item Name |
|---|---|---|---|
| $k_1$ | Transfer of information by an authorized process to an unauthorized process | Writing data by an authorized process to unauthorized addresses in RAM | $g_3k_1$ |
| $k_2$ | Acceptance of information from an authorized process by an unauthorized process | Writing data by an authorized process to unauthorized addresses in RAM | $g_3k_2$ |
| $k_3$ | Transmission of information through an unauthorized channel in an electromagnetic environment | Reading information using embedded malicious hardware | $g_3k_3$ |
| $k_4$ | Obtaining information from outside the authorized area | Retrieving from TEMPEST information about the elements of RAM | $g_3k_4$ |

**Table 5.** Typical threats to information flow $g_4 = (V_2, e_2, V_2)$.

| | Threat Description | Example of a Threat Implementation | Item Name |
|---|---|---|---|
| $k_1$ | Transfer of information by an authorized process to an unauthorized process | Receiving confidential information due to substitution of the address of the source process in RAM | $g_4k_1$ |
| $k_2$ | Acceptance of information from an authorized process by an unauthorized process | Receiving confidential information due to substitution of the address of the source process in RAM | $g_4k_2$ |
| $k_3$ | Transmission of information through an unauthorized channel in a virtual environment | Substitution of virtual address space | $g_4k_3$ |
| $k_4$ | Obtaining information from outside the authorized area | Retrieving from TEMPEST information about the elements of RAM | $g_4k_4$ |

**Table 6.** Typical threats to information flow $g_5 = (V_2, e_3, V_2)$.

| | Threat Description | Example of a Threat Implementation | Item Name |
|---|---|---|---|
| $k_1$ | Remote transfer of information by an authorized process to an unauthorized process | Covert redirection of information to an unauthorized network node | $g_5k_1$ |
| $k_2$ | Remote acceptance of information from an authorized process by an unauthorized process | Covert redirection of information to an unauthorized network node | $g_5k_2$ |
| $k_3$ | Transmission of information through a remote unauthorized channel in an electromagnetic environment | Driver substitution or installation of a malicious code as a result of unauthorized flashing of the Ethernet controller | $g_5k_3$ |
| $k_4$ | Obtaining information from outside the authorized area | Retrieving from TEMPEST information about the transmission channel | $g_5k_4$ |

**Table 7.** Typical threats to information flow $g_6 = (V_2, e_4, V_2)$.

| | Threat Description | Example of a Threat Implementation | Item Name |
|---|---|---|---|
| $k_1$ | Remote transfer of information by an authorized process to an unauthorized process | Covert redirection of information to an unauthorized address | $g_6 k_1$ |
| $k_2$ | Remote acceptance of information from an authorized process by an unauthorized process | Covert redirection of information to an unauthorized address | $g_6 k_2$ |
| $k_3$ | Transmission of information through a remote unauthorized channel in a virtual environment | Using an unauthorized network card driver and/or unauthorized protocol | $g_6 k_3$ |
| $k_4$ | Obtaining information from outside the authorized area | Network traffic analysis or network packets interception | $g_6 k_4$ |

**Table 8.** Typical threats to information flow $g_7 = (V_2, e_1, V_3)$.

| | Threat Description | Example of a Threat Implementation | Item Name |
|---|---|---|---|
| $k_1$ | Writing by an authorized process of information into an unauthorized data carrier | Unauthorized copying of a file | $g_7 k_1$ |
| $k_2$ | Reading by an unauthorized process of information from an authorized data carrier | Writing information to a file, access to which is not delimited (unauthorized file) | $g_7 k_2$ |
| $k_3$ | Transmission of information through an unauthorized channel in an electromagnetic environment | Using an unauthorized or compromised hard disk controller driver | $g_7 k_3$ |
| $k_4$ | Obtaining information from outside the authorized area | Retrieving from TEMPEST information about elements of hardware interfaces for connection and operation of input/output devices; setting embedded malicious hardware | $g_7 k_4$ |

**Table 9.** Typical threats to information flow $g_8 = (V_2, e_2, V_3)$.

| | Threat Description | Example of a Threat Implementation | Item Name |
|---|---|---|---|
| $k_1$ | Writing by an authorized process of information into an unauthorized data carrier | Writing protected information to an unauthorized (unprotected) file | $g_8 k_1$ |
| $k_2$ | Reading by an unauthorized process of information from an authorized data carrier | Unauthorized reading of a protected file | $g_8 k_2$ |
| $k_3$ | Transmission of information through an unauthorized channel in a virtual environment | Transfer of information using an unauthorized and compromised driver | $g_8 k_3$ |
| $k_4$ | Obtaining information from outside the authorized area | Reading residual information from virtual memory | $g_8 k_4$ |

Thus, a list of 32 typical threats to the confidentiality of information processed in a computer system was compiled. It is necessary to clarify once again that this list is not a complete list of threats:

— first, this study only addresses privacy threats;
— secondly, given the fact that technologies are developing at an increasing pace, we cannot accurately predict which input/output, storage or transmission devices will exist in a few years, let alone defining a complete list of threats to information that will be processed with the help of now defunct devices.

With all of this, we can say with confidence that the set of typical threats will remain unchanged, since the apparatus used in the basis of the threat model has a high degree of abstraction and is based on graph theory, and not on objects of the real world. Within the framework of the model, any device can be represented as an information transmission channel, regardless of its implementation. The specialist is only required to "not forget" about this channel (device) at the moment of describing the entire system. The implemented abstraction allows you to describe the system down to the minimum level

of interaction between elements. The specialist determines the depth of the detailed description of the system all alone, depending on the feasibility and requirements. To clarify, at this stage, we are not trying to fully automate the threat identification process. The developed model identifies only typical threats for a set of flows, i.e., a system. Identification of actual threats is another task, which is the next step (or the one after building a model of the violator).

To clarify, let's get back again to the example of sending an email and let's apply the proposed model to one of the flows. Consider only one flow $s_1 = (v_{1,1}, e_1, v_{2,1})$, remembering that it belongs to the class of typical flows $g_1$, and therefore it is necessary to analyze the following threats: $g_1k_1$, $g_1k_2$, $g_1k_3$, $g_1k_4$.

In the case of a $g_1k_1$ threat, the $v_{1,1}$ user is substituted by an unauthorized $v_{1,1}$ * user; as a result, this element can gain access to confidential information. An example of the realization of a threat is the transfer of a phone to a third party. An unauthorized user can get acquainted with the information in the information transmission channel (typed text in the input interface) or, using the mail client, read other letters sent or received earlier.

In case of a $g_1k_2$ threat, the $v_{2,1}$ mail client is replaced by the unauthorized $v_{2,1}$ * software; as a result, an authorized $v_{1,1}$ user can transfer confidential information to an unauthorized software tool. An example would be installing an application from an unverified source.

In case of a $g_1k_3$ threat, the $e_1$ communication channel is replaced. In this case, the communication channel is the I/O device, which is most likely a touch screen. An example of the implementation is the substitution of an input/output device: during the repair, an additional touch panel could be imposed on the touchscreen, similar to overhead keyboards for ATMs.

In case of a $g_1k_4$ threat, there is no direct intervention in the information channel, there is no substitution of any of the vertices or channel. Information is accessed from outside. An example of an implementation is the installation of a hardware or software backdoor. Indeed, the hardware backdoor does not replace any of the elements, it does not even interfere with their normal operation. Moreover, remote peeping can serve as an example of implementation, no one element of the elementary information flow has been compromised, and there is still information leakage outside the system.

Based on everything stated earlier, let us analyze the most popular analogue—the STRIDE methodology invented by Loren Kohnfelder and Praerit Garg [37]. The first thing that needs to be pointed out directly is the misuse of terms in this methodology. The name of the methodology itself is an abbreviation of the first letters of the names of threats: Spoofing, Tampering, Repudiation, Information disclosure, Denial of Service, Elevation of Privilege. The problem is that in the strict sense, these terms are not threats, but attacks. Moreover, each of them is aimed at a certain aspect of information security; however, the aspects described in the methodology expand the classic CIA triad, but do not coincide with the Parkerian Hexad. Instead of that, STRIDE has its own set of aspects, that fact indicates to the sophistication of the model.

In STRIDE, there is no division of threats by object of influence and there is no description of typical threats. Let us go back to our email transmission example. Which object of the system is the Information disclosure threat applicable to? To the channels? To the storage? To the information security tools? This definition is very general, which is why an information security specialist is forced to determine himself which nodes of the system to apply each of the threats. However, this is not a STRIDE problem. STRIDE does not have its own mechanism for describing the information system as a protected object. Yes, it can be applied with DFD, but DFD has several disadvantages already described in the second paragraph of this work.

For example, interaction between a user and an email client using a mobile device. Which part of a given local interaction system is each threat applicable to? In STRIDE, a problem comes to the fore, which is solved in the model proposed by the authors—the lack of typing and formalization. The specialist is not provided with a tool according to which he must describe the system in order to determine the information threats.

## 6. Conclusions

In the course of this study, two models were developed that have practical applicability in information security processes:

— a model of information flows;
— a model of information confidentiality threats.

The information flow model makes it possible to reduce the description of any information system to a limited set of elementary information flows. This model has a high level of abstraction, which makes it possible to describe any system regardless of its subject area.

The proposed model of information threats, based on the above-mentioned model of information flows, also has a high level of abstraction inherited from the basic model. The threat model contains 32 typical threats to information confidentiality.

Additionally, the proposed model differs from analogues in the following:

— the set of typical threats is finite;
— threats are clearly classified according to the object of access;
— separation of information flows into a discrete element of the system;
— the presence of a list of examples of the implementation of threats, the expansion of which will not affect the quality of the model;
— strict separation of threats and attacks.

**Author Contributions:** E.N.S., K.A.A. and S.A.A. wrote the paper. All authors have read and agreed to the published version of the manuscript.

**Funding:** This research received no external funding.

**Acknowledgments:** This research was funded by the Ministry of Science and Higher Education of Russia, Government Order for 2020–2022, project no. FEWM-2020-0037 (TUSUR).

**Conflicts of Interest:** The authors declare that they have no competing interest.

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
