# Peer review of "A Model of Threats to the Confidentiality of Information Processed in Cyberspace Based on the Information Flows Model"

_symmetry, doi:10.3390/sym12111840_

Round 1

Reviewer 1 Report

This isn't bad.  Some formatting oddness (unsure why you're using a dash as a bullet; your figures are fuzzy), but its readable and presented well.

Minor gripes:

  • Several key references are missing. Odd that you talk about STRIDE and threat modelling but don't mention Adam Shostack's work (who was hugely influential on the development of both), the relevant NIST standards should be talked about too.
  • 4 self citations is borderline inappropriate.

Bigger gripes:

  • Why is this going to a journal on Symmetry?  There isn't anything here about symmetry.  Wouldn't a cybersecurity venue be more appropriate?
  • Your conclusions are wrong.  You complain that STRIDE isn't doesn't coincide with the Parkerian Hexad—but why should it?  There is no consensus that it really brings much beyond the well established CIA model.  You also moan that it isn't clear what STRIDE threats are applicable to.  This is deliberate, and the intention is for the information security specialist to use their own judgement as to what aspects of a system need protection.  The purpose of the framework isn't to state in some formal notation exactly what must happen at every stage, but to frame and give structure to "the human in the loop"'s thought processes and process when developing their threat model. 

    Trying to eradicate that human process, trying to automate the identification of threats in a purely mathematically is doomed to failure: its impossible to specify everything and a changing threat landscape means your model is out of date as soon as you start. (see Moritz Y Becker's PhD thesis for an example of just how information flows get out of control rapidly).

That said, your model (whilst not being particularly different) seems sane enough, and it's a good read.

Author Response

Hello!

Thank you very much for remarks.

If you do not mind, I decided two create one general article for to reviewers.

I tried to highlight all of my fixes and expect that you will find all responses for your questions in attachment file.

Reviewer 2 Report

The work presents an interesting approach to the computer threats. Undoubtedly, this is a very important topic, especially now that the internet is present in every place and on any device. Language is correct, figures and tables are readable and intuitive.

The STRIDE model presented at work promises to be very good. However, there is no direct use of the model, which is probably not possible at this stage of the study. The authors emphasize the high level of abstraction, which is true. It is difficult to propose new elements of the model at this level of abstraction. I think the direct use of the model will force its expansion.

  • Comments to which you should correct or add a few words:
    Please correct the summary. The summary currently describes the chapters (this may be at the end of the first chapter). The summary should contain a description of the studies, not a description of the structure of the article.
  • Please correct words: ofHidden Markov - a lack of space.
  • Captions of tables 2-9, there are large digits, there should be lower indices.

Author Response

(The authors gave the same response as above.)
